# Recent Advances in Development of Natural Cellulosic Non-Woven Scaffolds for Tissue Engineering

**DOI:** 10.3390/polym14081531

**Published:** 2022-04-09

**Authors:** Mohammad Reza Aghazadeh, Sheyda Delfanian, Pouria Aghakhani, Shahin Homaeigohar, Atefeh Alipour, Hosein Shahsavarani

**Affiliations:** 1Faculty of Chemical Engineering, Babol Noshirvani University of Technology, Babol 47148, Iran; m.aghazadeh@stu.nit.ac.ir (M.R.A.); s.delfanian@stu.nit.ac.ir (S.D.); 2Department of Cell and Molecular Sciences, Faculty of Life Science and Biotechnology, Shahid Beheshti University, Tehran 1983963113, Iran; pouriaaghakhani97@gmail.com; 3Laboratory of Regenerative Medicine and Biomedical Innovations, Pasteur Institute of Iran, Tehran 1316943551, Iran; 4School of Science & Engineering, University of Dundee, Dundee DD1 4HN, UK; shomaeigohar001@dundee.ac.uk; 5Department of Nanobiotechnology, Pasteur Institute of Iran, Tehran 1316943551, Iran

**Keywords:** natural cellulose, tissue engineering, bone regeneration, skin regeneration, cardiac regeneration, vascular regeneration, neural regeneration

## Abstract

In recent years, tissue engineering researchers have exploited a variety of biomaterials that can potentially mimic the extracellular matrix (ECM) for tissue regeneration. Natural cellulose, mainly obtained from bacterial (BC) and plant-based (PC) sources, can serve as a high-potential scaffold material for different regenerative purposes. Natural cellulose has drawn the attention of researchers due to its advantages over synthetic cellulose including its availability, cost effectiveness, perfusability, biocompatibility, negligible toxicity, mild immune response, and imitation of native tissues. In this article, we review recent in vivo and in vitro studies which aimed to assess the potential of natural cellulose for the purpose of soft (skin, heart, vein, nerve, etc.) and hard (bone and tooth) tissue engineering. Based on the current research progress report, it is sensible to conclude that this emerging field of study is yet to satisfy the clinical translation criteria, though reaching that level of application does not seem far-fetched.

## 1. Introduction

Despite several successful clinical transplantations of allogeneic tissues and organs to treat diseases caused by failed or defected organs or tissues, the shortage of allogeneic donors and medical complications have always been hindering factors against their wide applicability [1]. As a new field of bioengineering, tissue engineering has enabled the creation of various tissue substitutes by combining engineering, cell biology, and materials science, thereby imitating the structural and physiological characteristics of native tissues [2]. The organ shortage as a clinical emergency can be conceivably overcome through tissue engineering [1]**.**

Cellulose, one of the most exploited natural materials on the planet, has made its way through civilizations from the early ages to the present [3]. The contribution of this carbohydrate to human life has attracted researchers’ attention in various fields of science. From food packaging and energy supply to biomedical and therapeutic applications, it has served as a valuable biopolymer [4].

Biomaterials have been widely investigated for tissue regeneration, particularly as ECM-mimicking structures that are biocompatible, biodegradable, and non-cytotoxic. They are desired to be low cost with proper physicochemical properties and fabricated in a simple manner [5]. Possessing tunable structural and chemical properties, cellulose and its derivatives have found multiple uses in the biomedical field to treat and regenerate damaged tissues such as bone [6], skin [7], heart [8], cartilage [9], and blood vessels [10]. The highly organized structure of cellulose consisting of several glucose monomers can endure biodegradation in the absence of cellulolytic enzymes. Owing to its biocompatibility, water retention capacity and absorption, optical translucency, and chemo-mechanical properties, cellulose can play the role of the ECM and mimic the natural microenvironment of the human body, thereby supporting cell growth and tissue regeneration [8,11]. The structural characteristics of cellulose vary based on the assemblies of microfibrils and arrangements of the β-(1,4′)-D-glucopyranose monomers. The naturally occurring cellulose I, the native form of cellulose, is characterized by particular hydrogen bonds and van der Waals interactions of two co-existing crystal arrangements: cellulose I_α_ (triclinic) and cellulose I_β_ (monoclinic). Two main types of natural cellulose, namely, bacterial and plant-based cellulose, have been extensively studied for bone and skin tissue regeneration due to their biocompatibility, hydrophilicity, non-toxicity, and other native tissue-mimicking abilities [12,13]. Bone and skin tissues are not the only regenerative territories that natural cellulose has conquered. It also has been applied for other tissue engineering purposes such as neural, adipose, tendon, lung, and cardiac regeneration [14].

This review discusses the recent studies and developments of natural cellulose for bone and skin tissue regeneration, published since 2014. Additionally, the last section addresses the application of bacterial and plant-based cellulose in other tissue engineering areas.

## 2. Cellulose Sources

Generally, cellulose can be derived from nature or obtained as a synthetic material. The crystal structure of cellulose differs based on its synthetic or natural origin. In this regard, natural cellulose consists of the cellulose I crystal structure, while synthetic cellulose has the cellulose II and III configurations [15]. Figure 1 illustrates how various forms of crystalline arrangements transform to each other in cellulosic structures [12].

In the plant cell wall, microfibrils are around 5–50 nm in diameter and possess highly crystalline and amorphous regions [16]. In the case these fibrils are subjected to a proper combination of physical, chemical, and enzymatic treatments, isolated cellulose nanocrystals would form [17]. Acid hydrolysis is the most commonly used procedure to prepare these cellulosic nanostructures. Due to their relatively high surface area, low density, and biodegradability and the presence of hydroxyl groups allowing for surface modification, cellulose nanocrystals have been largely researched for tissue engineering from both industrial and academic viewpoints [16,18]. Since CNCs are derived via the aforementioned procedures, we do not classify them as nature-derived cellulose.

All cellulose-synthesizing organisms, including bacteria, algae, tunicates, and higher plants, contain cellulose synthase proteins, which catalyze glucan chain polymerization [19]. Despite the fact that catalytic domains of cellulose synthases are preserved in all cellulose-synthesizing organisms, the vast differences in the organisms’ lifestyles and the structure of the cellulose they produce suggest that regulatory proteins and underlying mechanisms of cellulose synthesis may have evolved independently [20]. Bacterial cellulose (BC) and PC are two main classes of cellulose that are used in tissue engineering scaffolds. BC is a biomaterial produced naturally by bacteria. It has a unique cellulose nanofiber-weaved 3D network structure that causes its exceptional mechanical qualities, water holding capacity, and suspension stability. It also has a high purity, a high degree of crystallinity, and excellent biocompatibility and biodegradability. Accordingly, BC has attracted significant attention from both academia and industry [21]. PC typically comprises impurities including hemicellulose, lignin, pectin, and other compounds, whereas BC is almost pure, possesses a significantly higher water content, and shows a notably improved tensile strength due to its longer chain length [12]. Figure 2 shows the process of developing natural-cellulose-based scaffolds for various applications in tissue engineering.

The native cellulose structure is composed of two crystalline forms of Iα and Iβ (Figure 3). Since natural cellulose is obtained from different origins, these crystalline forms are found in various proportions. Microalgal cellulose and BC possess a rich content of Iα, while animal and higher plant cellulose are rich in Iβ, which is thermodynamically more stable [22].

## 3. Natural Cellulose for Bone Tissue Engineering

Over two million bone grafts are used per year to treat bone injuries and defects, thus notably affecting patients’ welfare and life quality [23]. Despite the several disadvantages, administering bone grafts is still the standard strategy for reconstructing bone tissues in clinical practice [24]. Both allo- and autografting strategies might encounter multiple clinical concerns such as limited amounts of graft material, antigenicity, short-term viability, infection, and unpredicted graft resorption. However, recent considerations have overcome some of these complications at the expense of important graft properties such as osteoconductivity, osteoinductivity, and mechanical strength [25]. As shown in Table 1, remarkable progress has been achieved recently in bone tissue engineering that led to some efficient treatments promoting bone regeneration. In this regard, it is now possible to tailor the physicochemical characteristics of scaffolds using nanotechnology to mimic the native bone tissue behavior [24].

BC, an abundant, inexpensive, slowly degradable, and biocompatible biomaterial, possesses unique characteristics that have drawn the attention of bone tissue engineering researchers [26]. Proposing a cost-effective bone tissue engineering strategy, ECM-mimicking 3D macro/microporous-nanofibrous BC (mNBC) scaffolds were produced from *Komagataeibacter europaeus* SGP37 and prepared via the freeze drying method. Then, the acquired bioscaffolds were combined with osteoinductive low-dose BMP-2-primed murine mesenchymal stem cells. The primary studies on the interaction of the scaffold material with the unprimed C3H10T1/2 cells confirmed the scaffolds’ ability to provide proper cell adhesion, growth, and infiltration. For osteogenic studies, the cells were preconditioned with 50 ng/mL BMP-2 for 15 min and then were cultured on the mNBC scaffolds for up to 3 weeks. The results showed that the mNBC scaffolds could partially promote the mineralization of the cells. Additionally, the scaffolds seeded with the low-dose BMP-2-primed cells displayed remarkably improved bone matrix secretion and maturation compared to the unprimed cells, indicating the osseointegration of the developed material [27]. A novel 3D printed porous scaffold was constructed using a blend of polycaprolactone (PCL), gelatin (GEL), and BC reinforced with different concentrations of hydroxyapatite (HA) nanoparticles. To obtain an optimal pore size and a uniform blending ratio, four different composites were fabricated whose infill rates varied from fifty to eighty percent. The ideal pore size for a bone tissue-mimicking ECM was achieved at the 80% infill rate and resulted in a more than 90% uniformity ratio. BC-containing composites showed a lower tensile strength and higher cell viability compared to composites without BC. Furthermore, including 0.25% HA in the blends enhanced cell adhesion and viability compared to the other composites. The comparative studies on these printed scaffolds demonstrated their potential application as bone implants [28]. In another study, the effects of incorporating BC into hexagonal boron nitride (hBN)-reinforced polyvinyl alcohol (PVA) to fabricate 3D printed bone tissue engineering scaffolds was investigated. *Gluconacetobacter xylinus* was exploited to produce BC membranes, and after the purification and oven drying steps, a mixture of PVA/hBN was heated and employed in the 3D printing device in addition to a BC solution to prepare the scaffold. The results showed that although the tensile strength decreased in the composition containing 0.5 wt.% BC, cell viability and cell adhesion significantly increased. Moreover, DSC analysis confirmed that the crystalline structure of PVA was not disrupted by the additives [29]. A 3D porous microsphere was fabricated by Zhang et al. [30] using chemically synthesized collagen (COL) and BC with additional Bone morphogenetic protein 2 (BMP-2). This highly porous scaffold was shown to have a particle size ranging from 8 to 12 microns, a pore volume of 0.59 cm^3^/g, and an average pore diameter of 198.5 nm. The in vitro studies revealed that these scaffolds promoted the proliferation, adhesion, and osteogenic differentiation of mouse MC3T3-EL cells, thus offering a promising capacity as biocompatible 3D-COL/BC/BMP-2 microsphere-based scaffolds for bone tissue engineering [30]. Codreanu et al. [31] assessed the in vitro biocompatibility and in vivo osteoblast differentiating ability of BC-modified polyhydroxyalkanoate (PHB/BC) scaffolds. The scaffolds were prepared via the salt leaching technique, with melting carried out using a Brabender Plastograph, and then tributyl citrate (TBC) plasticizer and NaCl particles were subsequently added to each melted mixture. The in vitro assessment confirmed the biocompatibility and the supporting role of the PHB/BC scaffolds for the proliferation of the 3T3-L1 preadipocytes. Furthermore, BC contributed to the differentiation of osteoprogenitor cells to osteogenic cells in the initial stages. An in vivo study on a critical-size mouse calvarial defect showed an intensified OSX expression and enhanced ALP activity in the first four weeks after the implantation of the scaffolds. Thus, these results indicated that the BC-PHB scaffold could promote osteoblast differentiation in vivo and eventually induce new bone formation, as verified by X-ray and histology/histomorphometry analysis [31].

As a natural-cellulose-based material, decellularized cabbage (DCB) was investigated for bone tissue engineering by Salehi et al. [32]. They decellularized CB via an organ perfusion method using sodium dodecyl sulfate (SDS), Triton-X100, and sodium hypochlorite and then washed it using deionized water, normal hexane, and phosphate-buffered saline (PBS). The as-prepared DCB scaffold had an irregular geometry (Figure 4a–f), whereby the seeded cells were entrapped (Figure 4g–i). Both the BET and the tensile test indicated the potential of the DCB scaffold to mimic the spongy bone tissues. Moreover, in vitro studies showed a significant increase in ALP activity and mineralization of bone marrow-derived mesenchymal stem cells (BM-MSCs) cultured on the scaffold compared to BM-MSCs cultured on tissue culture plates (TCPs) as the control groups. Gene expression analysis implied a higher expression of osteocalcin, collagen-1 (Col-I), Runx2, and ALP in the cells cultured on DCB leaves compared to those cultured in Petri dishes. Conclusively, taking into account the osteogenic capacity and bone healing potential of the scaffold, DCB was proven to be a remarkable ECM-mimicking structure for bone tissue engineering applications [32].

Benefitting from the structural similarities of *Bambusa vulgaris* and native bone tissue, a bone scaffold was made from an oxidized decellularized bamboo stem. The plant stem was treated with different ratios of SDS, Triton X-100, and sodium hypochlorite for 24 h, 48 h, and 72 h in a shaker, followed by washing with distilled water and later by a lyophilization process. The decellularized bamboo stem was oxidized using sodium periodate to improve its biodegradability and biocompatibility [33]. Despite the lower compressive strength of the oxidized bamboo stem, it still remained within the compressive strength range of cancellous bone (1.5 to 45 MPa) [34]. Therefore, this material can be used for restoration of the damaged maxillo-facial and cranial bones that are exposed to non-significant load levels. The oxidized bamboo fiber was notably coated with sericin, and, as a result, cell adhesion increased compared to the non-oxidized bamboo fiber. In vitro studies revealed that filopodial extensions, cell viability, and ALP activity were higher in the oxidized groups than in the non-oxidized groups. For in vivo studies, the oxidized scaffold was implanted subcutaneously in a rat model. The results confirmed the improved angiogenesis, biocompatibility, and biodegradation of the scaffold, and thus its promising potential as a bone scaffold for non-load-bearing applications [33]. In another investigation by Salehi et al. [35], onion skin was treated through a decellularization process similar to that in their other study [36], and its mechanical and structural characteristics were analyzed. The decellularized onion skin was shown to possess an ordered rectangular geometry, interconnected pores, moderate roughness, and high tensile strength (Figure 5a–d). Moreover, a water contact angle test indicated that the decellularized scaffold was amphiphilic. The in vitro tests based on BM-MSCs confirmed that the decellularized onion skin provides a biocompatible and non-toxic environment for cell growth up to five days (Figure 5e–h). Furthermore, the ALP activity and calcium content in the cells present on the scaffold were shown to be higher than those in the control groups (i.e., BM-MSCs cultured in TCPs). Additionally, ALP, Runx-2, osteocalcin, and Col-I had a significantly higher expression in the BM-MSCs cultured on the decellularized onion skin than in the control cells in TCPs. All these results validated the positive influence of the decellularized onion scaffold on the osteogenic differentiation of stem cells [35]. 

**Table 1 polymers-14-01531-t001:** Some recent studies on natural cellulose scaffolds developed for bone tissue engineering.

Cellulose Source	Bioscaffold Platform	Achieved Results	Reference
BC	Three-dimensional macro/microporous-nanofibrous BC scaffold co-cultured with low-dose BMP-2-primed murine mesenchymal stem cells	Owing to the ECM-mimicking architecture, the scaffold provided an ideal environment for the proliferation, adhesion, and infiltration of osteoblast cells.	Dubey et al., 2021 [27]
BC	Three-dimensional printed porous composite scaffolds based on polycaprolactone/gelatin/BC/hydroxyapatite	The composite scaffolds induced promising osteoblast cell viability and adhesion. The pore size of the scaffolds was ideal for bone tissue substitution.	Cakmak et al., 2020 [28]
BC	Three-dimensional printed porous composite scaffolds composed of polyvinyl alcohol (PVA)/hexagonal boron nitride (hBN)/BC	The pore size and homogeneous structure of the scaffolds were desirable for bone tissue engineering. The addition of BC to the polymer blend resulted in a significant increase in human osteoblast cell viability.	Aki et al., 2020 [29]
BC	Multistage structural 3D porous microsphere composed of collagen/BC/BMP-2	The porous microspheres promoted osteoblast differentiation and thus can be used to repair injured bone tissues.	Zhang et al., 2020 [30]
BC	BC-reinforced polyhydroxybutyrate (PHB) scaffolds	PHB/BC scaffolds implanted in mice with a calvarial defect enhanced in vivo osteoblast differentiation and bone formation.	Codreanu et al., 2020 [31]
BC	Antibacterial nanocomposite bioscaffolds based on BC/β-glucan incorporating hydroxyapatite nanoparticles (n-HAp) and graphene oxide (GO)	The antibacterial activity was proved by Gram staining. In vitro study using an osteoblast cell line revealed better biocompatibility and cell proliferation and adhesion due to the uniform distribution of the pore size, surface roughness, spongy morphology, and enhanced mechanical properties.	Khan, Haider et al., 2021 [37]
BC	Nanocomposite scaffolds composed of BC, magnetite (Fe_3_O_4_), and hydroxyapatite (HA)	The supermagnetic nanocomposite scaffold exhibited a high porosity of 81.1% and mechanical properties similar to those of human cancellous/trabecular bone. Moreover, it supports osteoblast cell attachment and proliferation, making it a candidate for bone tissue engineering.	Torgbo et al., 2019 [38]
BC	Fisetin-loaded BC scaffold	In vitro studies based on BM-MSCs showed no cytotoxicity and an increase in cell viability. The gene expression assay indicated the osteogenic potential of the fisetin-loaded BC scaffold.	Kheiry et al., 2018 [39]
PC	Decellularized cabbage	On co-culturing BM-MSCs and the decellularized scaffold, the bone-related genes were significantly expressed, which is due to the rough surface and high specific surface area.	Salehi et al., 2021 [32]
PC	Decellularized and oxidized bamboo stem	The hydrophilicity of the scaffold was increased by the oxidation process. Plus, in vitro studies validated the improved MSC viability, adhesion, and osteogenic differentiation with the oxidized decellularized plant scaffold compared to the control groups.	Mohan et al., 2021 [33]
PC	Outermost skin of onion	The decellularized scaffold maintains a porous structure, moderate roughness, and a high tensile strength. The in vitro assessments proved the pro-osteogenic potential of the scaffold.	Salehi et al., 2021 [32]
PC	Collagen-coated decellularized red apple	The decellularized scaffold possesses a high porosity. In vitro studies indicated a higher bone formation potential for the scaffold.	Latour et al., 2020 [40]
PC	Decellularized spinach leaf	The surface topography and vasculature of the scaffold supported the attachment and proliferation of cultured BM-MSCs. The genes expressed during in vitro studies showed the pro-osteogenic nature of the scaffold due to its optimum surface composition, hydrophilicity, and high specific surface area.	Salehi et al., 2020 [36]
PC	Decellularized apple, broccoli, sweet pepper, and carrot	In vivo studies on a rat calvarial defect model showed facilitated bone mineralization in the presence of the decellularized plant scaffold.	Lee et al., 2019 [41]
PC	Poly-L-lysine-coated decellularized carrot	After plant decellularization, no cytotoxicity was shown in vitro. The decellularized plant-based scaffold supported MC3T3-E1 pre-osteoblast cells’ adhesion, proliferation, and osteogenic differentiation.	Contessi Negrini et al., 2020 [42]

## 4. Natural Cellulose for Skin Tissue Engineering

The skin, the largest organ of the human body, is composed of the dermis, epidermis, and subcutaneous tissue. The external layer is the epidermis that has no blood vessels and contains only keratinocytes. This layer is divided into five sublayers, namely, the stratum basale, stratum spinosum, stratum granulosum, stratum lucidum, and stratum corneum from the basal layer to the surface, respectively. Blood vessels and nerves pass through a thicker skin layer in the middle called the dermis, including the papillary and reticular dermis [43]. The reconstruction of the epidermis and dermis as two constituent layers of skin is the main challenge in skin tissue engineering [14]. The subcutaneous tissue, composed of adipocytes and collagen, is the deepest layer of the skin [43]. Clinically, treatment of surface skin defects is difficult to achieve and requires more sophisticated strategies. In this regard, skin tissue engineering offers new opportunities for scaffolding, wound dressing, and skin substitution [14].

Several biomaterials such as chitosan [44], silk fibroin [45], collagen [46], and hyaluronic acid [47] have been proposed for wound healing and skin regeneration. Cellulose, as the most abundant natural polymer, has also shown promising wound healing potential. For instance, a BC-ε-polylysine (ε-PL) composite was prepared for wound dressing. BC was crosslinked with polydopamine (PDA) through a 24 h self-polymerization process, followed by ε-PL treatment at different concentrations. As revealed by XRD, PDA and ε-PL did not generate a new crystalline phase, but, as seen in the morphological images (Figure 6a), the color of the membrane changed from white to black. Nevertheless, the ε-PL treatment showed no impact on the microstructure of the as-prepared scaffold. The zone of inhibition and colony-forming unit (CFU) assays along with the Live/Dead fluorescent staining method revealed a profound antibacterial activity of the wound dressing. Additionally, the functionalized dressing material exhibited proper cytocompatibility and hemocompatibility in vitro. In vivo testing based on Sprague Dawley rats with full-thickness wounds infected with S. aureus indicated improved wound closure and skin regeneration compared to the control groups which were not treated (Figure 6b). Therefore, the functionalized BC-based wound dressing was shown to be able to properly disinfect skin wounds and stimulate their healing in clinical applications [48].

In an interesting study, an all-natural injectable hydrogel composed of dialdehyde BC (DABC)-reinforced chitosan was prepared via a simple dispersion for wound dressing [49]. In this composition, DABC was meant to perform as a non-toxic crosslinker. The as-developed dressing material showed optimum mechanical properties, injectability, promising drug sustained release, and antibacterial activity. Moreover, in vitro tests using L929 fibroblast cells demonstrated the biocompatibility of the CS-DABC hydrogel [49]. The byproduct of symbiotic culture of yeast and bacteria in kombucha is BC pellicles [50]. Furthermore, a BC-based sustainable kombucha (KBC) sheet, recently developed by Pillai et al., was partially acid hydrolyzed, and its ability to serve as a 3D printed scaffold was determined [50]. In this regard, two acidic solutions, namely, sulfuric acid and hydrochloric acid, were employed to partially hydrolyze the obtained BC. Better extrusion conditions were achieved when 30% sulfuric acid was used. The acid-hydrolyzed KBC showed controllable mechanical properties and was successfully extruded to a multilayered 3D structure, implying its proper formability and printability. The seeding of human adult dermal fibroblast and MC3T3-E1 cells on the scaffolds certified their biocompatibility and non-toxicity in vitro. As a result, this 3D printed gel scaffold has a remarkable potential to be used in soft tissue engineering applications such as wound healing or skin regeneration [50]. In another investigation, lightweight aerogels were prepared based on BC. Ag/BC hydrogels were prepared using different concentrations of AgNO_3_, and BC/polyaniline hydrogels were formed in the presence of PEG and HCl solution followed by the incorporation of ammonium persulfate to form BC/Ag/polyaniline nanocomposite hydrogels. The elastic gel-like behavior and antibacterial activity of the as-prepared material were characterized. In vitro studies using mouse fibroblast cells demonstrated a decline in cell attachment but increased proliferation [51]. Moreover, via modifications using montmorillonite (MMT), BC nanocomposites were fabricated by Sajjad et al. [52] to assure skin tissue regeneration. Ca-MMT, Na-MMT, and Cu-MTT were synthesized via an ion exchange process. Then, BC-based nanocomposites were obtained by immersing BC into modified and non-modified MMT suspensions, followed by freeze drying. The modified MMT-BCs imparted remarkable antimicrobial, wound healing, and regenerative capabilities in animal models. Given the flexibility of BC, burned skin of the mobile parts of the body, including the knees and elbows, could be regenerated using this promising scaffold.

Moreover, a technique for decellularizing tobacco BY-2 and rice cells and tobacco root tissues was introduced using deoxyribonuclease I (DNase I) without surfactants [53]. The THP-1-derived macrophage activation experiment indicated that the BY-2 cell-derived matrices did not stimulate TNF-α secretion. Thus, implementing the decellularized matrices into scaffolds can potentially avoid inflammatory responses. Human foreskin fibroblasts (hFFs) were used for in vitro assessment of decellularized BY-2 cell-derived matrices. The hFFs were shown to properly attach and grow on the plant cells and tissue-derived matrices. Thus, the cultured PC matrices were confirmed to be a promising platform for tissue engineering applications [53]. In a fascinating approach, *Borassus flabellifer* (BF) endosperms were decellularized to mimic the ECM for the purpose of tissue engineering. After decellularization, cellulose/chitosan (CS/CHI) hybrid scaffolds were obtained by treating decellularized endosperms. The as-prepared scaffolds showed improved thermal stability and compressive strength but a reduction in biodegradability compared to BF endosperms without SDS treatment as the control group. Furthermore, in vitro investigations using fibroblasts indicated that a better microenvironment is provided by the scaffolds, enabling enhanced cell adhesion and colonization. Taking into account such merits, the CS/CHI scaffold can be regarded as a promising candidate for tissue engineering and 3D cell culture [54].

## 5. Natural Cellulose for Cardiac Tissue Engineering

As one of the leading causes of death, cardiovascular disease is threatening human health across the globe. Heart transplantation is still the gold standard treatment for end-stage cardiovascular diseases [55]. Nevertheless, due to the limited number of heart donors, new reliable regenerative methods such as cardiac tissue engineering are highly demanded [56].

Exploiting the structural similarities of the vascular architecture in plant and animal tissues, the cardiac tissue engineering potential of fibronectin-coated decellularized spinach and *Artemisia annua* leaves, parsley stems, and peanut hairy roots was investigated extensively. Spinach is a highly available leaf model with extensive vascularity that could support the flow of human blood cell-like particles and human cell recellularization. Furthermore, when the plant scaffold was recellularized with human endothelial cells (HUVEC), the adhered cells were aligned on the inner side of the vascular structures. Other in vitro studies revealed the adhesion of human pluripotent stem cell-derived cardiomyocytes (hPSCMs) on the outer surface of scaffolds, demonstrating spontaneous contractions and promising calcium handling capacities after 21 days [57]. In another investigation, a comparative study was carried out between cardiac patches made of non-coated and collagen IV- or fibronectin-coated decellularized spinach aiming to address the inadequate vascularization typically found in tissue-engineered cardiac patches. Human-induced pluripotent stem cell-derived cardiomyocytes (hiPS-CMs) were used for in vitro studies on the decellularized scaffolds. Comparing the ECM protein-coated scaffolds with non-coated scaffolds, no improvement in cardiomyocyte cell adhesion, behavior, and contractility was observed. Therefore, it was suggested in this work that ECM protein coatings are not essential for creating cardiac patches from decellularized leaves. In fact, a stand-alone decellularized leaf patch can serve as a potential candidate to re-functionalize diseased cardiac tissue [58]. In a preclinical investigation by Simeoni et al. [59], autologous BM-MSCs were seeded on BC membranes, developing a cell delivery patch. To evaluate the tissue regenerative capacity of the developed patches, the researchers implanted them into the infarcted area of myocardial infarcted mouse models. After 14 days of incubation, the patch induced angiogenesis, reflected in the remarkable cell proliferation. Additionally, SEM analysis proved that the BC membrane could support the adhesion of the co-cultured cells. These data suggest the cardioprotective and cardioregenerative potentials of this natural-cellulose-based scaffold.

## 6. Natural Cellulose for Vascular Tissue Engineering

Recently, the applicability of decellularized leaves and onion skin as vascular patches was investigated thoroughly by Bai et al. [60]. The drug delivery potential of these structures was also assessed in this attempt. First, the leaves and onion skin were incubated in 10% SDS and then washed with PBS. The leaves were then bleached in a 10% sodium chlorite solution for 12 h and then washed with PBS to remove the bleaching agent (Figure 7a,b) and rhodamine water perfusion were tested in both leaves and stem (Figure 7c). Polylactic-co-glycolic acid (PLGA)-based rapamycin nanoparticles were fabricated (Figure 7d) and hydrogel obtained (Figure 7e). later perfused in the leaf vasculature, and coated on onion cellulose by immersing the hyaluronic acid-coated samples in a PLGA-based rapamycin nanoparticle suspension. Figure 8a–d demonstrates onion skin related camera and SEM images from the decellularization and its cellulosic fibers to the nanoparticle coated-ones. The as-prepared scaffolds were examined in a rat inferior vena cava patch venoplasty model. The implantation of the patch resulted in the rapid formation of thinner neointima. As a venous neointimal hyperplasia inhibitor, the administered nanoparticles formed thinner neointima for both decellularized scaffolds [60]. Based on this work, Bai et al. [60] recommended the plant grafts as potential scaffolds for human vascular implantations.

In another work, BC/potato starch (PS) membranes were developed by Liu et al. [61] by adding PS granules to the synthesis medium of BC, and then vascular-like tubes were formed via manual curling. The lower and upper layers of the as-synthesized membrane were used as the outer layer of the vascular graft and as the inner lumen, respectively. The 2 wt.% BC/PS grafts showed an oriented microporous outer structure and a dense inner surface and offered high patency of 75%. Furthermore, in vivo assessments on rabbit models revealed that the scaffolds rapidly promoted blood vessel regeneration. Such a hierarchical bioscaffold can be regarded as a promising artificial small-diameter vascular graft.

In a preclinical study, a tubular shape-memory BC membrane was developed and assessed in vitro and in vivo. This multilayered cellulosic structure was laden with endothelial, muscle, and fibroblast cells via microfluidics-based patterning and was shown to be able to imitate blood vessels in vitro. The in vivo implantation of this scaffold into the carotid artery of a rabbit revealed thrombus-free patency after 21 days [62]. In an in vivo assessment by Scherner et al. [63], BC grafts with a fiber structure were implanted in sheep models, thereby confirming the neoformation of a three-layered vascular wall without prothrombogenic or inflammatory signs.

## 7. Natural Cellulose for Dental Tissue Engineering

Dental complications including cavities, periodontitis, apical periodontitis, and pulpitis are among the costliest healthcare concerns. In addition to the economic burden both patients and healthcare systems encounter, many of these complications lead to tooth loss. As an alternative method to traditional tooth removal treatments and hard restorative mimetic materials, scaffold-based tissue engineering is being developed as a safe and practical regenerative therapy [64]. For instance, a mineral binder powder and a BC composite were synthesized by Voicu et al. [65] for application in endodontics. The mineral silicate cement binder was composed of SiO_2_, Al_2_O_3_, CaO, and ZnO at different concentrations and synthesized via a sol–gel method. After converting BC to a white-beige powder, it was mixed with a silicate cement powder to improve its binding properties, including its setting time and mechanical characteristics as well as biological features. Such a hydro compound showed an increased crystallinity degree and favorable adhesion to the surface of teeth. Moreover, in vivo studies revealed the ability of this structure to promote cell viability and proliferation while showing no cytotoxicity. Above all, it caused a significant mineralization level, indicating the high potential of this material for dental tissue engineering. In another study by An et al. [66], BC membranes were irradiated at 100 kGy and 300 kGy and used for guided bone regeneration in dental applications. In vitro and in vivo studies showed the higher cell viability and biodegradability of such membranes.

## 8. Natural Cellulose for Other Tissue Engineering Applications

Plant-based and BC scaffolds have been used for the regeneration of other tissues such as nervous system, adipose, and connective tissues. For instance, recently, Guo et al. [67] fabricated a BC-graphene foam (3D-BC/G) for neural stem cell (NSC)-based therapy. The 3D-BC/G foam was fabricated through culturing *A. xylinumon* on multilayer 3D graphene foams created via interfacial polymerization on the skeleton of synthetic foams. The structural and morphological analyses showed an increase in surface area, and many oxygens bearing functional groups, allowing for enhanced cell–scaffold interactions. In vitro studies validated the biocompatibility of the 3D-BC/G scaffolds when NSCs were cultured on them for three days. Further in vitro experiments indicated that the scaffolds promoted the growth, proliferation, and adhesion of NSCs and maintained their stemness. Altogether, the 3D-BC/G scaffold can serve as a promising conductive structure for neural tissue engineering.

To provide a structure for maintaining the patency of vascular networks, Zhang et al. [68] synthesized a BC (RMBC) membrane decorated with RGD peptide-grafted magnetic nanoparticles. Sequentially, a BC membrane was cultured by fermentation. Then, PEG-coated iron oxide nanoparticles were deposited on it through the Steglich esterification reaction, and finally, the magnetic BC was conjugated with RGD peptides. Culturing murine endothelial C166 cells on the as-fabricated membrane demonstrated better cell adhesion and proliferation than the stand-alone BC and magnetic BC (MBC). Meanwhile, applying a slow frequency using an external oscillating magnetic field promoted cell attachment so that endothelialization was achieved after culturing for four days. As suggested by the authors, the developed method is a non-invasive and convenient way to regulate endothelialization and can be used for vascular tissue engineering applications.

In an assessment attempt by Zhang et al. [69], the in vitro and in vivo biocompatibilities of BC were analyzed for corneal stroma replacement. Rabbit corneal epithelial and stromal cells were employed for an in vitro study that confirmed the biocompatibility and non-cytotoxicity of BC plus the creation of a favorable environment for the growth and adhesion of both cells. Slit-lamp examination and HE staining showed that BC preserved its transparency with no neovascularization and inflammation. Furthermore, an in vivo investigation showed no edema or inflammation in the cornea during the 90-day post-operation period. The researchers concluded that BC is a fascinating biomaterial for corneal stromal tissue engineering.

Binnetoglu et al. [70] conducted in vivo studies on 40 female Sprague Dawley rats to assess the potential of BC as a tubularizing biomaterial for repairing transected facial nerves. Whisker movement analysis and electrophysiological tests confirmed that, despite the lack of improvement in facial nerve function in the presence of the BC tubes, the regenerating axons were significantly higher in number than with other control methods which had not received facial nerve anastomosis after transection. This observation suggests that the BC hollow structure guides the fibers from the proximal to the distal nerve stump, enhances axonal regeneration, and provides a nerve conduit for neural regeneration [70]. In innovative research, an electrically conductive, biocompatible BC/polyaniline/nano-clay aerogel was fabricated via in situ polymerization of aniline on a BC/clay mixture. The aerogel showed a balanced level of thermal stability, biocompatibility, and flexibility. Furthermore, superior cell adhesion and proliferation were observed in vitro. Altogether, the acquired aerogel was demonstrated to be a remarkable candidate for neural regeneration [71].

In a multi-sectional investigation by Contessi Negrini et al. [42], decellularized apple- and celery-derived scaffolds were assessed to regenerate adipose tissue and tendons, respectively. After decellularization and washing using SDS, CaCl2, and distilled water, decellularized plants’ mechanical and morphological properties were examined. Apple-derived scaffolds possessing large and homogenous porosity were shown to be adequately fit for adipose tissue engineering applications. For the in vitro assessment, a preadipogenic cell line (3T3-L1) was cultured on the decellularized apple scaffold for 14 days. The results showed an increase in the metabolic activity of the cells as a proof for the scaffold-supported preadipocyte cell growth and proliferation followed by induced differentiation after seven days of culture. The obtained results demonstrated the applicability of the decellularized apple-derived scaffolds for adipose tissue regeneration. Furthermore, the differentiated adipocytes were more prominent in size and rounder in shape than those cultured in the preadipogenic medium. The celery-derived scaffolds were characterized as morphologically oriented structures with parallel pores, suitable for mimicking the native anisotropic connective tissue of tendons. The mechanical and biological tests indicated the potential of decellularized celery-derived scaffolds with regard to anisotropic tissue regeneration [42].

## 9. Conclusions and Future Outlooks

Over the last two decades, an expanding interest has evolved in the application of BC and PC for the development of tissue engineering scaffolds. This review aimed to offer a general perspective on the capacity of natural cellulose for tissue engineering purposes including tissue regeneration and healing. As we showed here, recent studies have proven that various cellulose-based compositions present favorable properties and thus functions that can promote cellular activities towards tissue regeneration. In this regard, a brief summary of well-designed 3D cellulosic nanocomposites and functionalized cellulosic biomaterials developed as porous tissue engineering scaffolds was discussed. Such scaffolds have been shown to stimulate vascularization and tissue neoformation and thus replace damaged or diseased bone, skin, corneal stroma, and heart and nerve tissues, among others. Among the mentioned target tissues, the bone and skin are the most studied. There is a high therapeutic potential of cellulosic scaffolds for other less investigated tissues, and this should be seriously taken into account. In the case of cellulose-reinforced composites, surface functionalization and thereby the homogeneous distribution of the cellulose phase within polymeric matrices (in particular the nonpolar ones) are still regarded as crucial challenges.

As an advanced class of cellulose-based materials for tissue engineering, cellulose hydrogel composites have drawn extensive attention. In this regard, an evolving interest has emerged with regard to the contribution of mechanotransduction to cell spread and cell differentiation. The cellulose phase could affect the design and structural properties of hydrogel nanocomposites, allowing for customization of the physical cues inside the neighboring ECM, thereby enhancing stem cells’ lineage-specific differentiation. On the other hand, a wide range of stimulus (e.g., temperature, pH, biomolecules, and ionic strength)-responsive cellulose-incorporated nanocomposite hydrogels can be developed in a controlled manner for tissue engineering purposes [16]. Since this provides a general overview on all classes of naturally derived cellulosic scaffolds, a whole other review article should be devoted to natural-cellulose-based hydrogels.

Despite the reported high quality of PC and BC scaffolds, some shortcomings hinder their applications in human tissue regeneration. The absence of antimicrobial, antioxidant, and stand-alone regenerative activity would necessitate related modifications by employing additives which induce the desired characteristics [72]. For non-biodegradable biomaterials, in applications that require in vivo degradation, it is proposed that this drawback be managed by various chemical modifications or treatments, such as oxidization. Thermal stability is another criterion which not only affects the processing stage of the scaffold but also correlates with its in vivo performance and properties, including hydrolytic degradation, and oxidation/crystallization. Additionally, when dry heat is applied in sterilization methods, the structure, morphology, and working and setting times during application vary based on the thermal stability. The low temperature stability of cellulosic structures has been addressed by the reinforcement of several inorganic nanoparticles, metal oxides, and nanotubes which possess high thermal stability due to their high crystallinity [73]. Although several practices have shown remarkable outcomes, most of them have resulted in uncontrollable degradation and inadequate biocompatibility [7]; thus, robust modification strategies for the fabrication of natural-cellulose-based structures with the desired features for human tissue regeneration are yet to be developed.

## Figures and Tables

**Figure 1 polymers-14-01531-f001:**
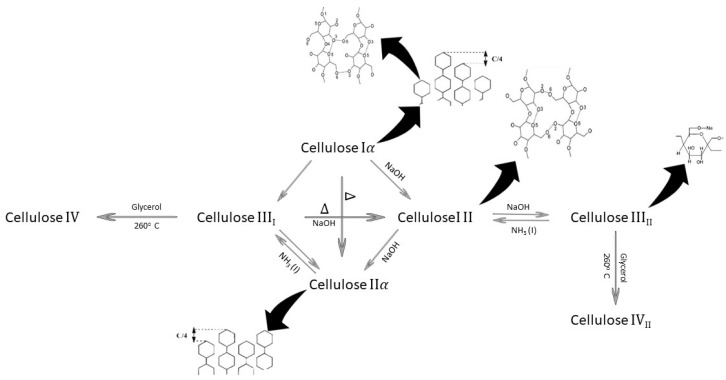
Scheme of the transformation cycle of cellulose crystals [12].

**Figure 2 polymers-14-01531-f002:**
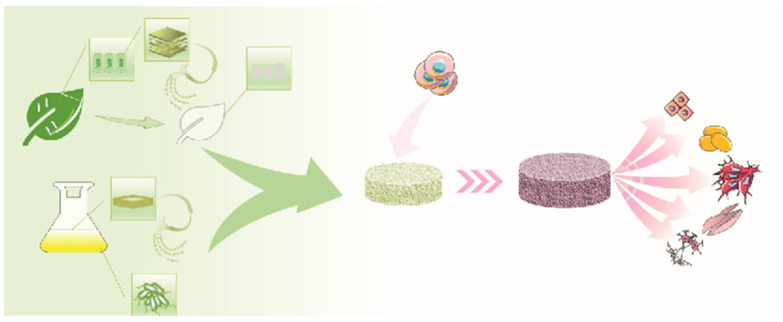
Schematic illustration of the fabrication procedure of natural cellulosic scaffolds: 1. bacterial inoculation or plant decellularization; 2. purification and slicing; 3. chemical/physical modifications; 4. culturing cells on the surface of the scaffold (2D cell culture) or 3D cell culturing for in vitro assessments.

**Figure 3 polymers-14-01531-f003:**
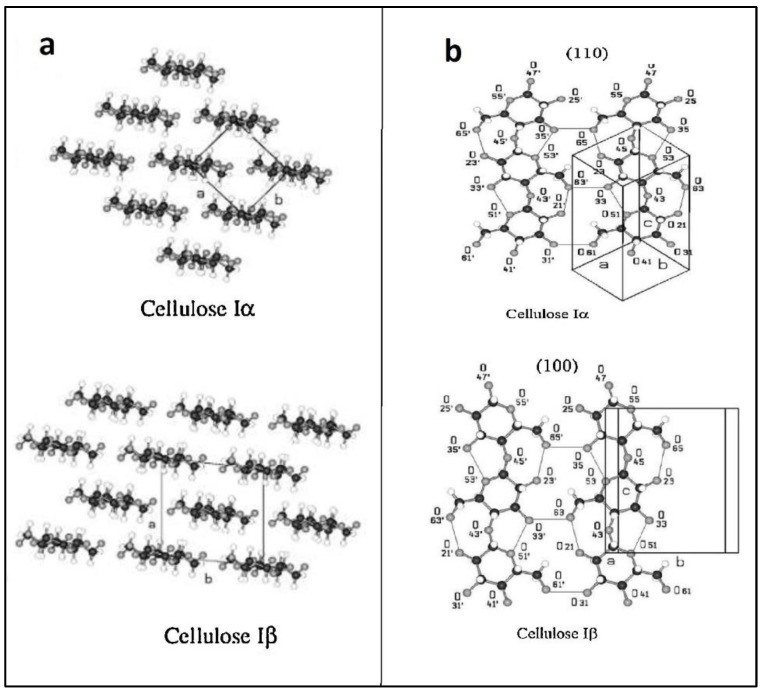
(**a**) Projections of the cellulose I crystal packings. (**b**) Two single sheets of cellulose Iα and cellulose Iβ on the (100) and (110) crystallographic planes, respectively [22].

**Figure 4 polymers-14-01531-f004:**
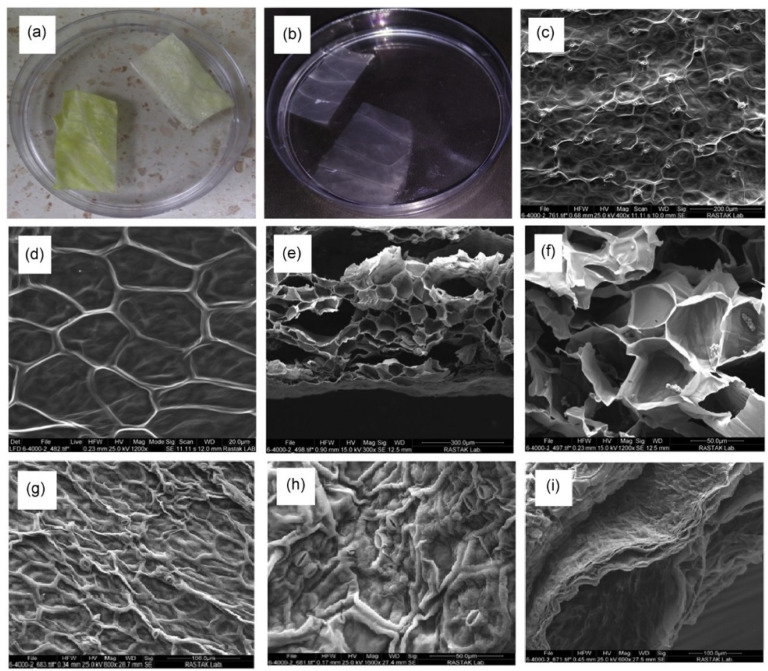
The camera images show the leaves of cabbage (Cb) (**a**) and decellularized Cb (**b**). (**c**,**d**) SEM images of decellularized Cb. (**e**,**f**) Cross-sectional SEM images of decellularized Cb. (**g**,**h**) SEM images of decellularized Cb cultured with BM-MSCs. (**i**) Cross-sectional SEM image of decellularized Cb cultured with BM-MSCs [32].

**Figure 5 polymers-14-01531-f005:**
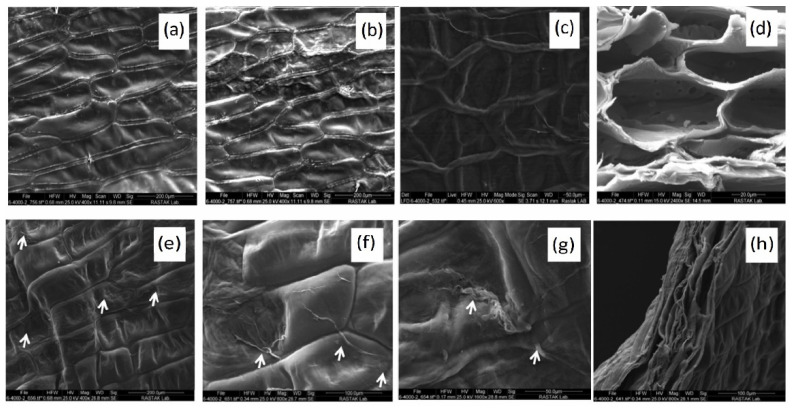
SEM images of the onion skin before (**a**) and after decellularization (**b**,**c**). (**d**) Cross-sectional SEM image of the decellularized onion skin after decellularization. SEM images of the surface (**e**–**g**) and cross-section (**h**) of the decellularized onion skin after 18 days of BM-MSC culture (the arrows mark the adhesion points of the cells on the surface) [32].

**Figure 6 polymers-14-01531-f006:**
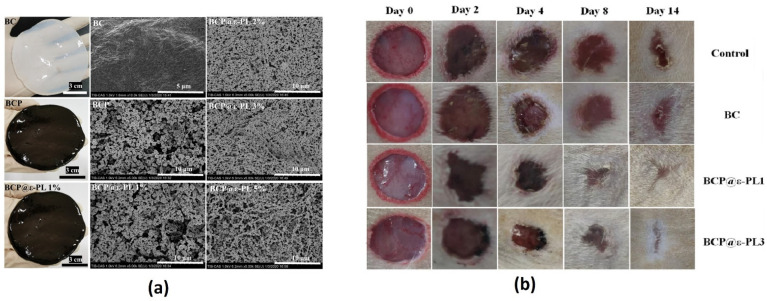
(**a**) Camera images of wound dressings made of BC, BC/PDA (BCP), and BC/PDA with 1% ε-PL and their corresponding SEM micrographs (note that the last (right) column contains SEM images of BCP@ɛ-PL dressings with higher additive concentrations). (**b**) In vivo wound healing potential of the BC and BCP@ ɛ-PL1&3 dressings compared to the control, imaged at different time points up to two weeks [48].

**Figure 7 polymers-14-01531-f007:**
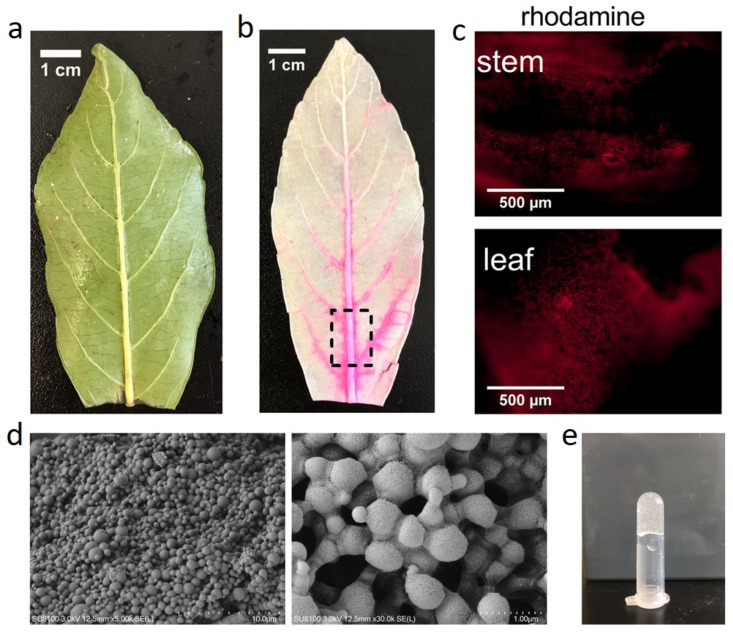
(**a**,**b**) The camera image of a leaf before and after decellularization, respectively. (**c**) Immunofluorescence images of a stem and leaf after rhodamine water perfusion. (**d**) SEM images of PLGA-based rapamycin nanoparticles at different magnifications. (**e**) The solidified hydrogel [60].

**Figure 8 polymers-14-01531-f008:**
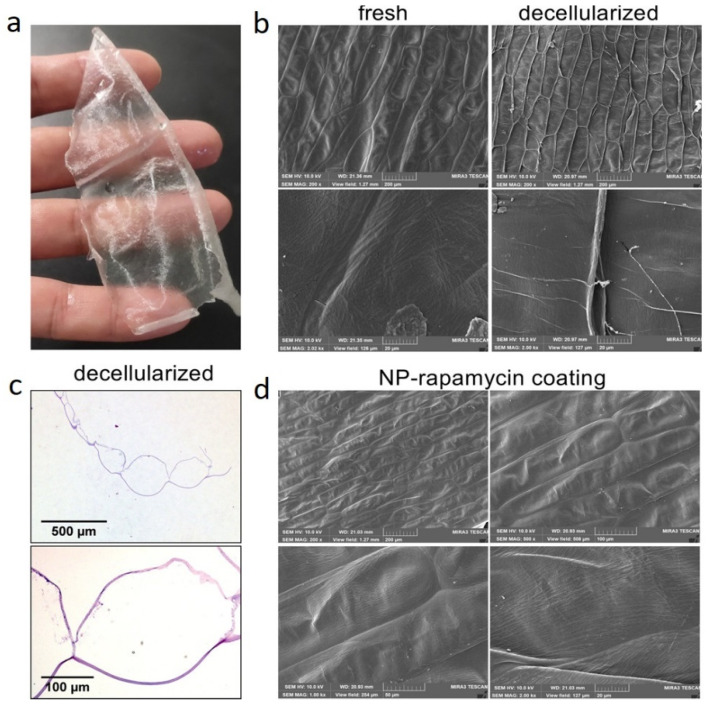
(**a**) Camera image of decellularized onion skin. (**b**) SEM images of onion skin before and after decellularization at different magnifications. (**c**) Microscale images of hematoxylin–eosin stained onion cellulosic fibers (**d**) SEM images of decellularized onion skin after being coated with rapamycin nanoparticles [60].

## Data Availability

No new data were created or analyzed in this study.

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
