# Peer review of "Recent Advances in Development of Natural Cellulosic Non-Woven Scaffolds for Tissue Engineering"

_polymers, 2022, doi:10.3390/polym14081531_

Round 1
Reviewer 1 Report
This review covers the advances generated in a very interesting line with potential growth, such as the use of natural cellulose for tissue engineering. So, I recommend its publication after considering the following comments:
-Avoid the use of the first person in the manuscript. Use passive sentences.
-A general conclusion or future perspective should be included in the abstract.
-What about the processing method of the materials? They could depend on the final application. Authors should include this in the review.
-A future perspectives section must be included.
Author Response
Dear Prof. Dr. Alexander Böker
Editor-in-Chief of Polymers
Attn: Manuscript ID polymers-1617549
Foremost, we would like to express our deep appreciation to you and reviewers for providing us with the insight and direction needed to complete our submitted manuscript under the title of “Recent Advances in Development of Natural Cellulosic Non-Woven Scaffolds for Tissue Engineering”.
Please kindly find the revised version of our manuscript in which the revised parts are indicated in red color based upon the replies to invaluable comments of reviewers.
We have carefully taken your comments and questions into the consideration for preparing revised manuscript. However, to some extent we would like to emphasize the response to your questions and major comments in this reply sheet as follows. I hope our responses to your comments meet your expectations. Please kindly let us know if you evaluate that this revision is enough mature to be accepted for publication in internationally renowned journal of Polymers.
Please kindly let us know if you or editors evaluate that this revision needs further revision. We would be very happy if our work could be acceptable for publication in internationally renowned journal of Polymers.
Reviewer #1: This review covers the advances generated in a very interesting line with potential growth, such as the use of natural cellulose for tissue engineering. So, I recommend its publication after considering the following comments.
Response to general comments of Reveiewer#1: We would like to take this opportunity to express our sincere thanks for your review and precious comments. In the revised version, we tried to explain further and improve the manuscript and discussion as indicated by Word’s tracker. We think that current review highlights the emerging natural-based cellulose scaffolds great potential in the field of tissue engineering. We have carefully taken your comments and questions into the consideration for preparing revised manuscript with further explanation as follows in detai.l
1. Avoid the use of the first person in the manuscript. Use passive sentences.
Thank you very much for take valuable time out of your schedule to read our manuscript. We rewritten a significant number of pharaghraphs and sentence, so most them have been changed to the passive voice.
2. A general conclusion or future perspective should be included in the abstract.
Thank you for your suggestion. In the current revised version, a conclusion is included in the abstract.
What about the processing method of the materials? They could depend on the final application. Authors should include this in the review.
Thank you very much for your time and invaluable suggestions. While the obtained charactristics of the scaffold structures are , indeed, directly impacted by the processing method, these procedures are mentioned in the revised manuscript.
- A future perspectives section must be included.
We thank the reviewer for this precious direction. The previous coclusion section were reconsidered to the conclusion and future prospectives, which comprises the major shortcomings of naturally-derived cellulosic scaffolds in contemporary regenerative applications, in addition to further suggestions to overcome these obstacle and recommeded ideas for future reviews in this field of research.
Reviewer 2 Report
In this review work authors presented the recent in vivo and in vitro studies on natural cellulose applied for soft (skin, heart, veins, nerve) and hard (bone and tooth) tissue engineering. It has been showed that natural cellulose, due to its biocompatibility, negligible toxicity and mild immune response, posses high potential as scaffold material for various regenerative aims.
The work was well-planned and gives the reader an outlook into interesting and up-to-date field of materials research. Some issues need however to be discussed in more depth:
- Cellulose nanocrystals, or CNCs, have recently emerged as a new class of versatile building blocks that offer high mechanical strength, low density, functionalisability, etc. Please complete your work with information on CNCs.
- Please add more information on cellulose scaffolds stability in vivo, e.g. hydrolysis effects;
- Recently, cellulose aerogels are vastly studied – please complete information on this class of materials;
- thermal stability of cellulose may hinder its processing with other polymers to develop new tissue engineering materials – please address this important issue;
- p. 13: „For instance, Voicu et al. [60] synthesized a mineral binder powder and BC-based composite for application in endodontics: - please specify the type of binder;
- Conclusions are rather poor, need to be rewritten.
Author Response
MAR 17, 2022
Dear Prof. Dr. Alexander Böker
Editor-in-Chief of Polymers
Attn: Manuscript ID polymers-1617549
Foremost, we would like to express our deep appreciation to you and reviewers for providing us with the insight and direction needed to complete our submitted manuscript under the title of “Recent Advances in Development of Natural Cellulosic Non-Woven Scaffolds for Tissue Engineering”.
Please kindly find the revised version of our manuscript in which the revised parts are indicated in red color based upon the replies to invaluable comments of reviewers.
We have carefully taken your comments and questions into the consideration for preparing revised manuscript. However, to some extent we would like to emphasize the response to your questions and major comments in this reply sheet as follows. I hope our responses to your comments meet your expectations. Please kindly let us know if you evaluate that this revision is enough mature to be accepted for publication in internationally renowned journal of Polymers.
Please kindly let us know if you or editors evaluate that this revision needs further revision. We would be very happy if our work could be acceptable for publication in an internationally renowned journal of Polymers.
Reviewer #2: In this review work authors presented the recent in vivo and in vitro studies on natural cellulose applied for soft (skin, heart, veins, nerve) and hard (bone and tooth) tissue engineering. It has been showed that natural cellulose, due to its biocompatibility, negligible toxicity and mild immune response, posses high potential as scaffold material for various regenerative aims.
The work was well-planned and gives the reader an outlook into interesting and up-to-date field of materials research. Some issues need however to be discussed in more depth.
- Cellulose nanocrystals, or CNCs, have recently emerged as a new class of versatile building blocks that offer high mechanical strength, low density, functionalisability, etc. Please complete your work with information on CNCs.
We thank the reviewer for this precious suggestion. Respecting the opinion of the reviewer, we have devoted a whole paragraph to cellulose nanocrystals. But based on the isolation methods of CNCs, we do not consider them to be classified as natural biomaterials though it is originally natural. In another word, we just considered scaffolds with less processing and chemical reconstruction as natural biomaterials (such as decellularized herbal tissues). Considering this definition, CNC is not categorized natural but can be considered natural-derived.
- Please add more information on cellulose scaffolds stability in vivo, e.g. hydrolysis effects;
Thank you for this comment. Through rewriting the conclusion section, we provided the main drawbacks of employing natural cellulosic scaffolds, including low thermal stability and non-biodegradability which is directly in correlation with the hydrolysis effect.
- Recently, cellulose aerogels are vastly studied – please complete information on this class of materials;
Thank you for this invaluable suggestion. Although in most studies, cellulosic aerogels have been fabricated based on unnatural sources, we referred and added those that were prepared based on natural ones.
- Thermal stability of cellulose may hinder its processing with other polymers to develop new tissue engineering materials – please address this important issue;
Respond to comment #2 of reviewer #2.
- 13: „For instance, Voicu et al. [60] synthesized a mineral binder powder and BC-based composite for application in endodontics: - please specify the type of binder;
Thank you kindly for this comment. We provided the requested information to the paragraph.
- Conclusions are rather poor, need to be rewritten.
We thank the reviewer for this precious direction. The previous conclusion section was reconsidered to the conclusion and future prospectives, which comprises the major shortcomings of naturally-derived cellulosic scaffolds in contemporary regenerative applications, in addition to further suggestions to overcome these obstacles and recommended ideas for future reviews in this field of research
Reviewer 3 Report
I think that the topic is of interest and worth to be published, but I have several points, which have to be "changed". A lot of relevant papers dealing with bacterial cellulose and "skin" are not mentioned and discussed. Please do so. What are the limitations of this review? What kind of results and studies are missing? What are the next steps? Please state and discuss!!
Author Response
Dear Prof. Dr. Alexander Böker
Editor-in-Chief of Polymers
Attn: Manuscript ID polymers-1617549
Foremost, we would like to express our deep appreciation to you and reviewers for providing us with the insight and direction needed to complete our submitted manuscript under the title of “Recent Advances in Development of Natural Cellulosic Non-Woven Scaffolds for Tissue Engineering”.
Please kindly find the revised version of our manuscript in which the revised parts are indicated in red color based upon the replies to invaluable comments of reviewers.
We have carefully taken your comments and questions into the consideration for preparing revised manuscript. However, to some extent we would like to emphasize the response to your questions and major comments in this reply sheet as follows. I hope our responses to your comments meet your expectations. Please kindly let us know if you evaluate that this revision is enough mature to be accepted for publication in the internationally renowned journal of Polymers.
Please kindly let us know if you or editors evaluate that this revision needs further revision. We would be very happy if our work could be acceptable for publication in internationally renowned journal of Polymers.
Reviewer #3: I think that the topic is of interest and worth to be published, but I have several points, which have to be "changed". A lot of relevant papers dealing with bacterial cellulose and "skin" are not mentioned and discussed. Please do so. What are the limitations of this review? What kind of results and studies is missing? What are the next steps? Please state and discuss!!
We do appreciate your time in reviewing our work and precious comments. According to your suggestion, we have added more relevant papers related to bacterial cellulose application in tissue engineering. The previous coclusion section were reconsidered to the conclusion and future prospectives, which comprises the major shortcomings of naturally-derived cellulosic scaffolds in contemporary regenerative applications, in addition to further suggestions to overcome these obstacle and recommended ideas for future reviews in this field of research.
Thanks in advance,
Hosein Shahsavarani, Ph.D.
Principal Investigator and Assistant professor
Laboratory of Regenerative Medicine & Biomedical Innovations
Pasteur institute of Iran
Email: hosein.shahsavarani@gmail.com; h_shahsavarani@pasteur.ac.ir

Round 2
Reviewer 1 Report
The authors have taken into account all the revierwers' comments, improving the quality of the review. So, I recommend its publication in the present form.
Reviewer 2 Report
Authors provided proper explanations to my comments, and the revised manuscript can be published.